# Nutritional Status and Physical Exercise Are Associated with Cognitive Function in Chinese Community-Dwelling Older Adults: The Role of Happiness

**DOI:** 10.3390/nu16020203

**Published:** 2024-01-08

**Authors:** Jianghong Liu, Michael Pan, McKenna Sun, Haoer Shi, Rui Feng

**Affiliations:** 1School of Nursing, University of Pennsylvania, Philadelphia, PA 19104, USA; pdyd@seas.upenn.edu (M.P.); haoershi@seas.upenn.edu (H.S.); 2School of Engineering and Applied Science, University of Pennsylvania, Philadelphia, PA 19104, USA; 3College of Arts & Sciences, University of Pennsylvania, Philadelphia, PA 19104, USA; 4Department of Biostatistics, Epidemiology, and Informatics, Perelman School of Medicine, University of Pennsylvania, Philadelphia, PA 19104, USA; ruifeng@pennmedicine.upenn.edu

**Keywords:** cognitive decline, physical exercise, nutritional status, happiness, aging

## Abstract

We aim to assess the relationship between nutrition status, physical exercise, and cognitive function and particularly examine how happiness modifies and mediates the relationship, among 699 seniors aged 60 and above in Shanghai, China. Linear regression models were used to validate the effects of nutrition and exercise on cognitive function and to test their interaction effects with happiness. When the interactions were significant, stratified analyses in sub-groups were conducted. Mediation effects of happiness were examined using two-step causal mediation models. We confirmed that better nutrition (*p* < 0.001) and exercise (*p* = 0.009) were significantly associated with less cognitive decline. Furthermore, the effects of nutrition and exercise on cognitive decline were significant in the unhappy (happiness < 20) (*p* < 0.001) and younger (age < 74) sub-groups (*p* = 0.015). Happiness partially mediated 11.5% of the negative association of cognitive decline with nutrition (*p* = 0.015) and 23.0% of that with exercise (*p* = 0.017). This study suggests that happiness moderates and partially mediates the effects of exercise and nutrition on cognitive status. The beneficial effects of exercise and nutrition were stronger in less happy or younger seniors. Future intervention studies are required to confirm this path relationship.

## 1. Introduction

The increasing challenge of age-associated cognitive decline has become a prominent global concern as the population ages. Modifiable lifestyle factors such as nutritional status and physical activity, especially exercise, have been recognized as protective factors against cognitive decline [1]. An extensive amount of literature has underscored the link between nutritional status and cognitive decline [2,3,4,5,6,7,8,9]. Both cross-sectional [2,3,4] and longitudinal [5] studies from various populations consistently reported correlations between malnutrition or nutritional risks and higher levels of cognitive decline, whereas the dietary intake of nutrient-rich foods including tea [6], egg [7], fruits, milk, and yogurt [8,9] is associated with reduced risks of cognitive decline. Furthermore, intervention studies, implementing diverse dietary patterns, have demonstrated efficacy in attenuating the progression of cognitive deterioration [1,10,11]. Adequate nutrition is proposed to be fundamental for sustaining optimal brain function [12,13], whereas deficiencies in essential nutrients contribute to cognitive impairment by affecting essential metabolic activities including enzymatic reactions and neurotrophic factor expressions [14,15].

Similarly, general physical activity, specifically exercise, emerges as another pivotal lifestyle factor significantly impacting cognitive function. Observational studies suggest that regular physical exercise corresponds to noticeable improvements in cognitive function [1,16,17], while physical inactivity is associated with cognitive deficits [18,19]. Intervention studies employing physical exercises of various types, including aerobic exercises [20,21], mind–body exercises [22,23], and culturally tailored exercises [24], have proven to be effective. Recent meta-analyses on both longitudinal studies [25] and randomized clinical trials (RCTs) [26] also highlighted reduced cognitive decline risks and improved cognitive function. Multifaceted explanations, encompassing improved blood flow, neurogenesis, and the release of neurotrophic factors, have been proposed to underlie this protective effect [27].

While both nutritional status and physical exercise have consistently shown associations with cognitive function, intervention studies examining the impact of both diet and physical activity on cognitive decline have yielded mixed results [1,28,29]. These inconclusive findings underscore the complexity of the relationship between lifestyle factors and cognitive health and the need to consider other potential factors influencing this relationship, such as happiness. Questions also remain regarding the underlying pathways through which nutrition and physical exercise impact cognition.

Happiness is increasingly recognized as a crucial factor for health conditions and has been associated with cognitive function [30,31]. Mounting evidence has also indicated a significant positive association between happiness and physical exercise [32,33,34]. However, few studies have explored its associations with nutritional status and its potential mediating effects on the link between physical activity and cognition. Furthermore, demographic factors such as age also showed influences on various aspects of older adult lifestyles [35,36], and may hence influence the link between nutrition, exercise, and cognition.

Therefore, the purpose of this study is firstly to assess the direct associations among nutritional status, physical exercise, and cognition. Next but importantly, we seek to explore potential modification and mediation effects of happiness in their relationship, in a sample of Chinese community-dwelling older adults. Lastly, we also explore the modifying roles of demographic variables such as age.

## 2. Materials and Methods

### 2.1. Study Population and Settings

The current study is part of a cross-sectional study on Chinese community-dwelling older adults. The cohort recruited a sample of 755 Chinese older adults (44.5% males and 55.5% females) aged over 60 years from multiple communities in the Shanghai metropolitan city in 2017. A comprehensive survey covering demographics, happiness, cognition, lifestyle, sleep, nutrition, and social connections was delivered. A total of 150 interviewers, comprising 40 nursing supervisors from Shanghai Longhua Hospital and 110 interns from Shanghai Shuguang Hospital, assisted with interview procedures. For older adults who were not capable of completing the survey independently, the interviewers verbally guided them through each item and recorded their results. The survey was completed by 709 participants, with 699 remaining in the present study after excluding 10 individuals who had incomplete cognition data. Informed consent was obtained from all participants and institutional review board approval was obtained from the ethics committees for research at both Shanghai Shuguang Hospital and Shanghai Longhua Hospital. All research was performed in accordance with the relevant guidelines and regulations.

### 2.2. Measures

#### 2.2.1. Nutritional Status

Nutritional status was evaluated using the Mini Nutritional Assessment (MNA^®^), a screening tool to assess nutritional status in older adults [37]. The instrument is composed of 18 questions, which are grouped into 4 sub-domains: anthropometric measurements, global assessment, dietary questionnaire, and subjective assessment [38]. The sum of the instrument score ranges from 0 to 30, with lower scores indicating risks of malnutrition. For linear regression analysis, the score was categorized into 3 groups using recommended thresholds: 1 = malnutrition (MNA < 17); 2 = at risk of malnutrition (17 ≤ MNA < 23.5); 3 = normal nutritional status (MNA ≥ 24). MNA was reported to show an internal consistency of 0.83 and a test–retest reliability of 0.89 [39]. The reliability of the Chinese version of the MNA was 0.698 [40]. In the present study, the reliability α was 0.633.

#### 2.2.2. Physical Exercise

Physical exercise was measured by the General Information Survey through the following question: “Do you often participate in physical exercise (such as Tai chi, jogging, walking, public square dancing, ball sports, biking, etc.)? 1 = no, 2 = yes.”.

#### 2.2.3. Cognitive Function

We assessed cognitive decline using the memory/cognition problems syndrome scale of the self-report Older Adult Self-Report (OASR) [41]. The OASR is a self-administered instrument examining various domains of older adults including adaptive functioning, personal strengths, behavioral, emotional, and social problems [41]. OASR has been proven to be valid for Chinese-speaking older adults [42,43], with the cognitive subscale being previously reported in the same population [44]. The cognitive decline scale was derived from 9 items, asking about cognitive issues including “cannot concentrate” and “forget names”. All items were rated on a three-point scale (0 = not true, 1 = sometimes true, and 2 = often true). The total score was calculated by summing 9 items and normalizing under the T-score format (mean = 50, SD = 10). Higher scores indicated a greater degree of cognitive decline.

#### 2.2.4. Happiness

Happiness was evaluated by the Subjective Happiness Scale (SHS) [45]. SHS is a 4-item instrument for the measurement of global subjective happiness [45]. The items ask if the respondent considered themself a happy person, happier than their peers, generally very happy, and generally not very happy. Each question is scored from 1 to 7, which sums to a total score of 4 to 28. Higher scores represent greater happiness. SHS has been reported to show good reliability (Cronbach’s alpha = 0.79 to 0.94) [45] and has been validated in a Chinese population [46]. In this study, a threshold of 20 was set to dichotomize participants into unhappy (SHS < 20) and happy (SHS ≥ 20) sub-groups.

#### 2.2.5. Covariates

Sociodemographic and other relevant information including age, gender, income, education, and marital status were used as covariates in the current study. Income was categorized into three groups: low income, middle income, and high income (i.e., <1000 RMB, 1000–3000 RMB, >3000 RMB, 1 RMB ≈ 0.15 USD in 2017). Education was grouped into below high school degree, high school degree, and college degree or above. Marital status was coded as living with spouses (married or cohabiting with spouses/partners), and others (unmarried, divorced, separated, or widowed).

### 2.3. Statistical Analysis

Basic demographic characteristics were summarized using descriptive statistics including mean, standard deviation (SD), and percentages as appropriate. To assess the marginal associations between nutritional status (indicated as nutrition below), physical activity (indicated as exercise below), and cognitive decline, Pearson’s correlation coefficients in-between and with continuous demographic variables were first calculated. The associations were then assessed using linear regression models, adjusted for age, gender, income, education, and marital status. Separate models were constructed for nutrition and exercise, with age, gender, income, education, and marital status adjusted. The interaction effect of nutrition and exercise was also examined. Due to the positive skewness of the cognitive decline score, log-transformation was applied as a sensitivity analysis. To examine the interactions with happiness and age, we added their cross-product with nutrition or exercise, respectively. If the interaction was significant, we conducted stratified analyses in dichotomized sub-groups, specifically in unhappy (happiness < 20) and happy (happiness ≥ 20) groups for the dissection of happiness interactions and in young–old (age < 74) and old–old (age ≥ 74) groups for that of age. Mediation analyses were then implemented using a two-step causal medication model if happiness mediated the association between the original nutrition score, exercise, and cognitive decline. A 1000 bootstrapping estimation was used, and 95% confidence intervals were reported. All analyses and result visualizations were performed using RStudio Version 4.3; *p*-values less than 0.05 were considered statistically significant.

## 3. Results

### 3.1. Sample Characteristics

The basic characteristics of the older adults are presented in Table 1. Of the 699 included participants, 312 (44.84%) were males, while the average age was 69.58 (SD = 7.46) years. A total of 428 (62.12%) older adults reported having below high school degrees, 182 reported having high school degrees (26.42%), and the remaining 79 (11.47%) of older adults had obtained college degrees or higher. For marital status, 499 (71.70%) reported living with their spouses. As for nutritional status (indicated as nutrition below), 25 (3.78%) reported having protein-calorie undernutrition, 271 (40.94%) reported being at risk of malnutrition, and 366 (55.29%) reported having normal nutrition. For physical exercise (indicated as exercise below), 477 (71.70%) participants reported doing exercise. The raw average score of cognitive decline was 3.02 ± 3.15. The average score for happiness was 17.77 (SD = 3.13, range 4 to 28).

Bivariate correlations between nutrition, exercise, cognitive decline, and happiness are summarized in Appendix A. Both nutrition (r = −0.25, *p* < 0.001) and exercise (r = −0.13, *p* < 0.001) were significantly associated with cognitive decline. Happiness significantly correlated with nutrition (r = 0.21, *p* < 0.001), exercise (r = 0.15, *p* < 0.001), and cognitive decline (r = −0.17, *p* < 0.001). In addition, higher ages corresponded to lower levels of nutrition (r = −0.13, *p* < 0.001), but not exercise and happiness (*p* > 0.05).

### 3.2. Nutrition, Exercise, and Cognitive Decline

We first examined the associations between nutrition, exercise, and cognitive decline, with results summarized in Figure 1 and Table 2. For nutritional status, both the malnutrition (β = 10.63, SE = 2.09, *p* < 0.001) and the at-risk levels (β = 3.10, SE = 0.78, *p* < 0.001) significantly predicted elevated cognitive decline compared to normal nutrition, after adjusting for covariates. For physical activity, exercise was associated with decreased cognitive decline with respect to non-exercise (β = −2.13, SE = 0.81, *p* = 0.009). However, the interaction of malnutrition and exercise showed no significance.

### 3.3. Interaction Effects of Happiness and Age

We then investigated whether the identified associations varied across different subpopulations. The interaction terms with happiness were added to previous models. Malnutrition corresponded to a higher cognitive decline score (β = 36.37, SE = 9.47, *p* < 0.001), while a higher happiness score predicted improved cognition (β = −0.43, SE = 0.16, *p* = 0.010). In addition, a significant interaction between happiness and malnutrition was observed (β = −1.79, SE = 0.60, *p* = 0.003). Similarly, exercise (β = −11.78, SE = 4.44, *p* = 0.008) and happiness (β = −0.86, SE = 0.19, *p* < 0.001) exhibited a significant negative association with cognitive decline, with their interaction also being significant (β = 0.58, SE = 0.25, *p* = 0.020). No three-way interaction between nutrition, exercise, and happiness was identified. In sensitivity analysis, no significant changes were identified with the log-transformation applied. To explore the identified interaction, participants were dichotomized into happy (happiness ≥ 20, *N* = 202) and unhappy (happiness < 20, *N* = 480) sub-groups (Table 2) where the analyses were replicated. Nutrition and exercise still significantly predicted an elevated cognitive decline level in the unhappy group (Figure 2). Specifically, malnutrition and at-risk levels exhibited an increase of 11.10 points (β = 11.10, SE = 2.30, *p* < 0.001) and 2.96 points (β = 2.96, SE = 0.95, *p* = 0.002) with reference to the normal level, respectively, while exercise corresponded to a decrease of 2.35 points (β = −2.35, SE = 0.96, *p* = 0.015). However, the associations in the happy group did not show a statistical significance. The interactions with various demographic variables including age, gender, and education level were also explored, with significant interactions only observed with age (β = −0.71, SE = 0.25, *p* = 0.006), and a significant malnutrition–cognition association, particularly in the young–old (age < 74) but not the old–old (age ≥ 74) sub-population (Appendix A).

### 3.4. Mediation Effects of Happiness

Given the significant associations between nutrition, exercise, happiness, and cognitive decline, we further investigated the potential mediating roles of happiness (Table 3). As shown in Figure 3, with nutrition exerting a negative total effect on cognitive decline (β = −0.78, *p* < 0.001), happiness partially mediated this association (indirect effect: β = −0.09, *p* = 0.006; direct effect: β = −0.69, *p* < 0.001), accounting for 11.54% of the total effect. Happiness also significantly mediated the association between exercise and cognitive decline (indirect effect: β = −0.48, *p* = 0.016, 22.97%). The direct effect of exercise on cognitive decline remained significant (β = −1.61, *p* = 0.050) and accounted for 77.03% of the total effect.

## 4. Discussion

As the global population skews towards an older demographic, it has become increasingly important to explore and implement interventions that target lifestyle choices as a method of preserving the healthy aging process. Particularly, nutrition and exercise emerge as two implementable strategies linked to cognitive decline [1]. This cross-sectional study involving Chinese community-dwelling older adults yielded several key findings. First, poor nutritional status correlates with a higher cognitive decline, while exercise is associated with a lower decline in cognitive function. Second, these associations were modified by happiness, proving significant specifically in the unhappy group. Finally, and most importantly, happiness serves as a mediator in the associations of nutrition and exercise with cognitive decline. Better nutrition and exercise correspond to a higher happiness level, which are further associated with decreased cognitive decline.

Our results indicated that both nutrition and physical exercise were individually associated with cognitive function, showing consistency with the literature. Previous research has extensively documented the association between poor nutrition and depressed cognitive function in elderly adults, particularly those with lower vitamin levels [47]. Among elderly individuals with mild cognitive impairment, a decline in plasma amino acid levels—derived from protein intake—correlates with an increased likelihood of developing Alzheimer’s disease (AD) [48]. Furthermore, numerous studies suggest that regular physical exercise and interventions involving exercise may improve cognitive function for both cognitively healthy individuals and those experiencing cognitive decline [49]. Primary mechanisms through which exercise exerts such an impact include the increased expression of brain-derived neurotrophic factors (BDNFs) and interleukin-6 (IL-6) in the hippocampus, subsequently enhancing memory pathways affected in cognitive decline [50]. Put together, our findings highlight the importance of nutrient supplementation and a broader intake of food groups and contribute to a clearer picture of exercise’s benefits as possible protectors against cognitive decline [1].

While a direct association was found in our study, we observed no interaction between nutrition and physical exercise on cognitive decline. To our knowledge, two prior studies have explored this synergistic effect, both aligning with our findings—Nijholt et al. reported no interaction between nutrition and physical activity in urban Dutch older adults, while Scarmeas et al. concluded that the combined effect of adhering to a Mediterranean diet and engaging in physical activity did not differ from their putative additive effect on AD risk [51,52]. Our study’s findings further contribute to the understanding of the relationship between these variables, suggesting that nutrition and exercise may individually contribute to cognitive function, but their coupled influence may be minimal.

Previous research has indicated that better nutrition and physical exercise are associated with increased life satisfaction and happiness in older adults [32]. However, the interplay between these factors and how happiness influences their dynamics has not been thoroughly explored. In the current study, happiness was found to modify the associations between nutrition, exercise, and cognitive decline. Specifically, the nutritional status and physical exercise of older adults experiencing unhappiness significantly predicted cognitive decline. This could be explained by the potential protective effects of happiness against the associations of malnutrition and a sedentary lifestyle with cognitive decline. Evidence suggests that happiness may enhance the supply of the neurotransmitter precursors to BDNFs [53,54,55], thereby attenuating the impact of malnutrition and the lack of exercise on cognitive function. These findings offer an additional outlet to develop feasible interventions for older adults’ mental health—improving their happiness may mitigate the negative impacts of poor nutrition and exercise.

Age also serves as a modifying factor in the relationship between nutrition, exercise, and cognition, with the findings being significant only in the young–old group (<74 years). Multiple existing studies have consistently reported a similar age stratification in their findings. Laukka et al., investigating the association between the rate of cognitive decline and various modifiable lifestyle factors, including diet and physical activity, discovered that the associations were significant solely in the young–old (<78 years) group [56]. Yoshimura et al. identified an association between being at risk of malnutrition and depression, a well-established risk factor for cognitive decline, in the young–old (<74 years) but not the old–old group [57]. Despite these findings, the underlying reasons for such age-specific phenomena remain elusive. One potential explanation is that the metabolic rate of the old–old group is relatively low, making them less affected by lifestyle factors [58]. This consequently leads to less pronounced associations between cognitive decline and nutrition or exercise. Our results highlight the preventive implications particularly to young–old adults and underscore the necessity of an in-depth assessment of age-specific dynamics within these associations.

Importantly, our results suggest that happiness is the key intermediate outcome for good lifestyle habits and cognitive function. Previous findings from Shi et al. indicated that physical activity mediated the positive correlation between happiness and cognitive function in Chinese middle-aged and older adult samples [31]. Our results from an exclusively older adult Chinese population expand upon these findings to show that happiness also operates as a mediator between physical exercise and cognitive function, accounting for 23% of the association. One potential explanation of these mediating effects may arise from the release of neurochemicals, such as endogenous opioids and endocannabinoids, accompanying physical exercise [59]. These chemicals may lead to a sense of euphoria, promoting both sustained or increased physical exercise and happiness in the long term [59]. Happiness-relevant neurotransmitters, such as dopamine and serotonin, may further be linked to cognition [60]. In the aging process, the decline of these two neurotransmitters is associated with cognitive deficits, reduced plasticity, and declining neurogenesis [61]. The maintenance of happiness holds the potential to counteract these effects. Given happiness’s crucial role as a mediator in the association between exercise and cognitive decline, targeting and enhancing older adults’ happiness may amplify the efficacy of their exercise in mitigating cognitive decline.

To our knowledge, no studies have examined the effect of happiness on the relationship between nutrition and cognitive health in older adults. Our study contributes a novel pathway where happiness mediates their relationship. Possible mechanisms for this impact can be inferred from the current literature. Specifically, a diverse array of fruits and vegetables are natural dietary sources of dopamine and serotonin, both of which play a crucial role in fostering positive mood and happiness [60,62]. Consuming a varied and nutritious diet may contribute to increased production of these neurotransmitters, creating a sustained state of happiness in older adults. Additionally, emerging research highlights the significance of the gut microbiome as a key link between diet and mood. Poor microbiome health, indicative of a suboptimal diet, has been associated with psychiatric disorders [63]. Gut microbiota, particularly *Bacillus* and *Lactobacillus* strains, act as major producers of biologically active dopamine and serotonin, respectively, which are then directly conveyed to the brain, impacting the mental state [62]. Consequently, detrimental changes to gut health via a poor diet may be implicated in negative changes in happiness and mood. Our study’s findings are pivotal in clarifying and expanding the role of happiness in older adults’ mental acuity, emphasizing the importance of developing interventions in this area to strengthen the impact of nutrition on mitigating cognitive decline.

These findings should be interpreted considering the limitations of this study. First, due to the cross-sectional nature of this study, it may not be possible to establish a causal relationship between the discussed variables. Second, the collected nutrition status data may not be fully comprehensive of each individual’s nutritional habits, because these data were based on a self-report instrument rather than a 24 h dietary record. This extends to the data collection on physical exercise as well—there was a lack of systematic data collection on physical exercise, and the self-report may not have captured all nuances in the adults’ exercise habits. Finally, since this research was conducted in community-dwelling adults from a major metropolitan city, these results may not be fully generalizable to the whole Chinese population.

## 5. Conclusions

In conclusion, this study revealed that nutrition and exercise as lifestyle factors were significantly linked with cognitive decline in Chinese community-dwelling older adults, where happiness plays both moderating and mediating role in these relationships. Our findings are crucial to informing feasible interventions for older adults to preserve their mental acuity in the aging process. These results hold major implications in providing accessible and feasible interventions for older adults. As the first study to connect happiness with nutrition and exercise in their associations with cognitive decline, this research may offer insights to inform the future design and implementation of cognitive interventions tailored to promoting positive emotional status and demographic sub-groups, ultimately contributing to the development of effective strategies for mitigating cognitive decline in aging populations. These findings may also encourage older adults to develop and maintain adequate lifestyle habits focusing on nutritional and physical exercise whilst pursuing activities that promote happiness.

## Figures and Tables

**Figure 1 nutrients-16-00203-f001:**
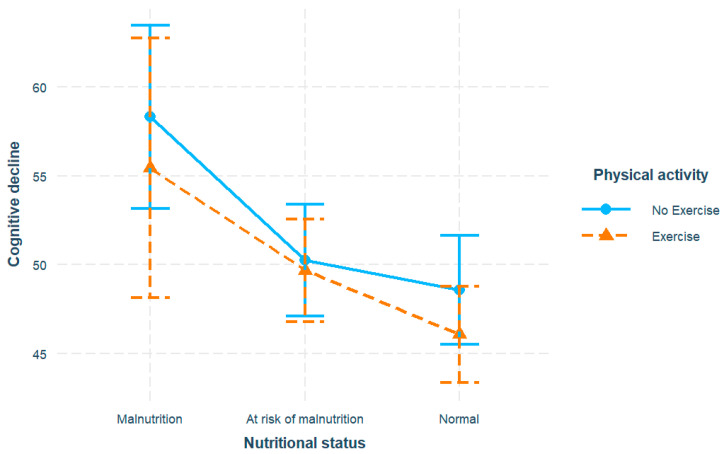
Association of cognitive decline with physical activity and nutritional status. Model adjusted for age, gender, income, education, marital status.

**Figure 2 nutrients-16-00203-f002:**
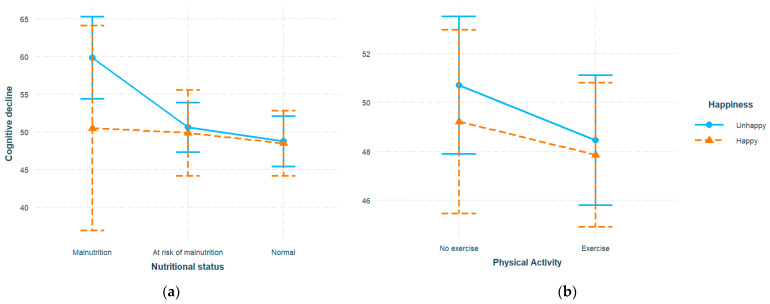
Stratified associations of cognitive decline across unhappy (happiness < 20) and happy (happiness ≥ 20) happiness sub-groups: (**a**) associations of cognitive decline with nutritional status; (**b**) associations of cognitive decline with physical activity. All models adjusted for age, gender, income, education, marital status.

**Figure 3 nutrients-16-00203-f003:**
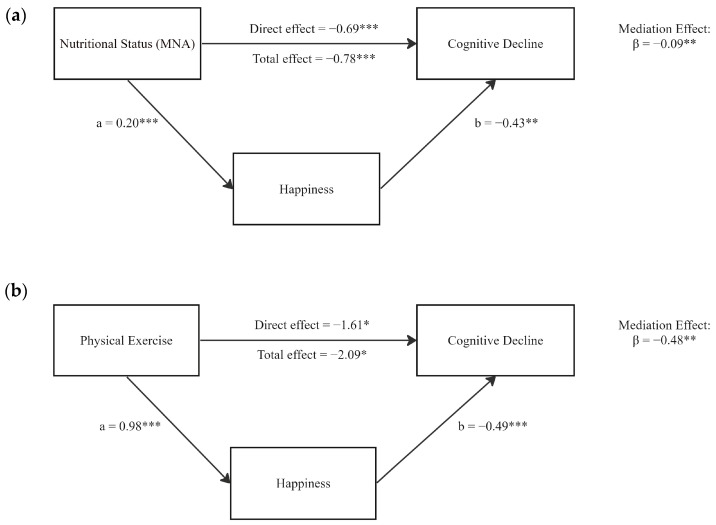
Mediation effects of happiness on associations between nutrition, exercise, and cognitive decline. (**a**) Mediation of happiness on Nutrition-Cognition effects; (**b**) Mediation of happiness on Exercise-Cognition effects. Mediation analyses were performed with 1000 bias-corrected bootstrapped samples. Mediation model adjusted for age, gender, income, education, and marital status. β, estimated regression coefficient; SE, standard error; *: *p*-value < 0.05, **: *p*-value < 0.01, ***: *p*-value < 0.001.

**Table 1 nutrients-16-00203-t001:** Basic characteristics of participants.

	Overall, *N* = 699
Age, mean (SD)	69.58 (7.46)
Age group, *N* (%)	
Young-old (age < 74)	524 (75.18)
Old-old (age ≥ 74)	173 (24.82)
Gender, *N* (%)	
Male	313 (44.84)
Female	385 (55.24)
Income, *N* (%)	
Low (<1000)	72 (10.42)
Middle (1000–3000)	270 (39.07)
High (>3000)	349 (50.51)
Education, *N* (%)	
Below High School	428 (62.12)
High School	182 (26.42)
College degree or above	79 (11.47)
Marital status, *N* (%)	
Living with spouse	500 (71.74)
Other	197 (28.26)
Nutrition score, mean (SD)	23.47 (3.38)
Nutritional status	
Malnutrition	25 (3.78)
At risk of malnutrition	271 (40.94)
Normal	366 (55.29)
Physical activity, *N* (%)	
Exercise	477 (68.73)
No	217 (31.27)
Happiness, mean (SD)	17.77 (3.13)
Happiness group, *N* (%)	
Happy (happiness ≥ 20)	202 (29.62)
Unhappy (happiness < 20)	480 (70.38)
Cognitive decline, mean (SD)	3.02 (3.15)

Notes: SD, standard deviation. Sum across nominal variables may not add to total number, due to missing data. Cognitive decline scores are presented as raw scores.

**Table 2 nutrients-16-00203-t002:** Adjusted association of cognitive decline with nutritional status, physical activity, stratified by happiness.

Predictors	Overall (*N* = 699)	Unhappy Group (*N* = 480)	Happy Group (*N* = 202)
	β (SE)	β (SE)	β (SE)
Nutrition (Normal as reference)			
At-risk	3.10 (0.78) ***	2.96 (0.95) **	2.90 (1.53) †
Malnutrition	10.63 (2.09) ***	11.10 (2.30) ***	9.05 (5.61)
Exercise (No as reference)			
Yes	−2.13 (0.81) **	−2.35 (0.96) *	−1.16 (1.64)

Note: Nutrition and exercise associations estimated from separate models. All models adjusted for age, gender, income, education, marital status. Β, estimated regression coefficient; SE, standard error; *: *p*-value < 0.05, **: *p*-value < 0.01, ***: *p*-value < 0.001. † indicates marginal significance.

**Table 3 nutrients-16-00203-t003:** Mediation results of happiness on nutrition, exercise, and cognitive function.

Paths	Difference (95% CI)	Percentage %	*p*-Value
Nutrition ⟶ Cognition			
Total Effect	−0.78 (−1.00, −0.55)	100	<0.001
Direct Effect	−0.69 (−0.92, −0.46)	88.46	<0.001
Indirect Effect (Nutrition → Happiness → Cognition)	−0.09 (−0.16, −0.03)	11.54	0.006
Exercise ⟶ Cognition			
Total Effect	−2.09 (−3.70, −0.48)	100	0.011
Direct Effect	−1.61 (−3.22, −0.003)	77.03	0.050
Indirect Effect (Exercise → Happiness → Cognition)	−0.48 (−0.94, −0.15)	22.97	0.016

Note: Mediation analyses were performed with 1000 bias-corrected bootstrapped samples. Mediation model adjusted for age, gender, income, education, and marital status.

## Data Availability

The data presented in this study are available on request from the corresponding author. The data are not publicly available due to privacy.

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
