# Peer review of "Nutritional Status and Physical Exercise Are Associated with Cognitive Function in Chinese Community-Dwelling Older Adults: The Role of Happiness"

_nutrients, 2024, doi:10.3390/nu16020203_

Round 1

Reviewer 1 Report

Comments and Suggestions for Authors

Summary statement:

The authors investigated the relation among nutritional status, physical activity, and cognitive function in a Chinese population. It's a promising study, but some points need to be resolved.

Major recommendations

The manuscript has several orthographical errors. For this reason, I suggest a reviser to check the correct orthography.

Physical exercise and physical activity are not synonymous. I recommend specifying and using the most appropriate term for your study.

The keywords used in the abstract are not Mesh´s. We suggest verifying.

The manuscript constantly presents statements without citations (introduction and discussion). I suggest that the authors carefully check the writing of the manuscript and use the appropriate citations and references.

The title suggests that the study will investigate the effects of nutritional status and physical activity on cognitive function, but this is not clear in the objective. It is essential to reformulate the introduction and the objective of the study so that it is clear

In characterizing the participant population, has a sample calculation been carried out to determine the ideal number of participants in the study? The authors need to present this information.

The resolution of the figures in the graphs is extremely low, it is crucial to substitute them with images of the quality required for an article. Additionally, what software was used to create the figures? Provide this information.

The authors constantly repeat the citations used in the introduction in the discussion section. This is not recommended. The authors are expected to present a summary of the study's main findings, discuss the reasons for these outcomes with the existing literature, and present the limitations and future perspectives of their study. In addition, the discussion is presented in topics, which is unusual. The discussion section needs to be completely changed.

Minor recommendations

In "Table 1. Basic characteristics of participants." Data on nutritional status, physical activity, happiness, and cognitive decline are essential to the study. I recommend that they be presented in graphs rather than tables.

It is recommended that authors provide the DOI of the studies in the references. I suggest adding.

Author Response

For research article

Response to Reviewer X Comments

1. Summary

Thank you very much for taking the time to review this manuscript. Please find the detailed responses below and the corresponding revisions highlighted in yellow the re-submitted files.

2. Questions for General Evaluation

Reviewer’s Evaluation

Response and Revisions

Does the introduction provide sufficient background and include all relevant references?

Can be improved

Please see below.

Are all the cited references relevant to the research?

Can be improved

Please see below.

Is the research design appropriate?

Yes

Thank you.

Are the methods adequately described?

Can be improved

Please see below.

Are the results clearly presented?

Can be improved

Please see below.

Are the conclusions supported by the results?

Yes

Thank you.

3. Point-by-point response to Comments and Suggestions for Authors

The authors investigated the relation among nutritional status, physical activity, and cognitive function in a Chinese population. It's a promising study, but some points need to be resolved.

Comments 1: The manuscript has several orthographical errors. For this reason, I suggest a reviser to check the correct orthography.

Response 1: Thank you for pointing this out. We have checked the writing thoroughly. Please find corrections highlighted.

Comments 2: Physical exercise and physical activity are not synonymous. I recommend specifying and using the most appropriate term for your study.

Response 2: We certainly agree with your point. Given that our survey question is mainly about exercises, we have decided to change the wording to physical exercise. This term has been updated throughout the manuscript accordingly. Please see the title, keywords, page 1 line 33, page 2 line 45, 56, among others.

Comments 3: The keywords used in the abstract are not Mesh´s. We suggest verifying.

Response 3: Thank you for your kind suggestion. Despite not being MeSH terms, the keywords cognitive decline and physical exercise are entry terms of MeSH terms Cognitive Dysfunction [ID: D060825] and Exercises [ID: D015444], respectively, and would be linked with MeSH terms in searching. In addition, the keywords are more specific to the topic of the paper and are consistent with the terms used throughout the manuscript. Therefore, these two keywords are kept. The remaining keywords Nutritional Status [ID: D009752], Happiness [ID: D006240], and Aging [ID: D000375] are all MeSH Terms and are not changed as well.

Comments 4: The manuscript constantly presents statements without citations (introduction and discussion). I suggest that the authors carefully check the writing of the manuscript and use the appropriate citations and references.

Response 4: Thanks for commenting on this. We have carefully checked the introduction and discussion section to ensure proper evidence support and citation for each statement. More literature was synthesized, with new references, including [20,22,23] and [47-64], added. We have also moved the citations close to their most relevant text to improve readability. Please see page 1 lines 34-35, page 2 lines 49-53, and the discussion section.

Comments 5: The title suggests that the study will investigate the effects of nutritional status and physical activity on cognitive function, but this is not clear in the objective. It is essential to reformulate the introduction and the objective of the study so that it is clear

Response 5: Thank you for mentioning this. While investigating these effects, we focused specifically on the role of happiness on them. We have now revised the objective in the abstract (page 1 lines 14-15) and also the last paragraph of the introduction (page 2 lines 70-74) to clarify our objective. The introduction section was also reorganized to make the structure clearer, where the first two paragraphs summarized the associations of nutrition and exercise with cognitive function, respectively, paragraph 3 discussed potential gaps, and paragraph 4 introduced the concept of happiness (page 2 lines 63-69).

Comments 6: In characterizing the participant population, has a sample calculation been carried out to determine the ideal number of participants in the study? The authors need to present this information.

Response 6: Thank you for this helpful suggestion. Unfortunately, we did not calculate the ideal sample size for this study in advance, and it is inappropriate to calculate that after data has been collected. However, the current sample size of around 700 is comparable to or exceeds studies investigating similar topics, employing cross-sectional design, and/or utilizing the same instruments. For instance, [1,2] using the same nutrition instrument, [3] using the same happiness instrument, and [4] using OASR, the cognition instrument. We believe this could to some extent support that our sample size is adequate for obtaining meaningful and statistically significant results.

Comments 7: The resolution of the figures in the graphs is extremely low, it is crucial to substitute them with images of the quality required for an article. Additionally, what software was used to create the figures? Provide this information.

Response 7: Thank you for mentioning this. We believe the figure resolution issue is potentially due to inappropriate file conversion from word to pdf file. We would confirm with the journal editor regarding this issue. The figures were created using R. This information is now included in the Methods Section, page 4 line 156, “All analyses and result visualization were performed using RStudio Version 4.3.”

Comments 8: The authors constantly repeat the citations used in the introduction in the discussion section. This is not recommended. The authors are expected to present a summary of the study's main findings, discuss the reasons for these outcomes with the existing literature, and present the limitations and future perspectives of their study. In addition, the discussion is presented in topics, which is unusual. The discussion section needs to be completely changed.

Response 8: Thank you for your suggestion. We have addressed each of your above concerns. 1) We have included a broader set of citations in addition to references in the introduction section, please see references [47-64]; 2) We have reorganized the contents in the discussion section to include summary of main findings, in-depth comparison with existing literature, potential mechanisms underlying findings, and finally implications and limitations; 3) We have removed the sub-headings in the discussion section. Overall, we have made thorough revision throughout the discussion section, including literature synthesis, logical flow, and exploration of key mechanisms to support our findings, please see highlighted in the Discussion section.

Comments 9: In "Table 1. Basic characteristics of participants." Data on nutritional status, physical activity, happiness, and cognitive decline are essential to the study. I recommend that they be presented in graphs rather than tables.

Response 9: This is a really great suggestion. However, we believe it is not easy to illustrate these data using figures given our condition. There are both continuous and categorical outcomes, whose scales vary significantly. This could lead to mixed presentation of the data in figures. In addition, there are no substantial extra information provided by just presenting a single mean/SD value or count of participants in figures. Therefore, we have decided to keep the current presentation.

Comments 10: It is recommended that authors provide the DOI of the studies in the references. I suggest adding.

Response 10: Thanks for mentioning. We have now added DOI for most references, except for books and instrument manuals for which DOI is unavailable (please see highlighted in the References).

4. Additional clarifications

To potential editor reviewing this, the reviewer has mentioned that the figure resolution is extremely low. We would like to confirm if it was an issue due to file conversion, or do we need to provide figures of higher resolution. Thank you!

References

  1. Feng, L.; Chu, Z.; Quan, X.; Zhang, Y.; Yuan, W.; Yao, Y.; Zhao, Y.; Fu, S. Malnutrition is positively associated with cognitive decline in centenarians and oldest-old adults: A cross-sectional study. eClinicalMedicine 2022, 47, doi:10.1016/j.eclinm.2022.101336.
  2. Boquete-Pumar, C.; Álvarez-Salvago, F.; Martínez-Amat, A.; Molina-García, C.; De Diego-Moreno, M.; Jiménez-García, J.D. Influence of Nutritional Status and Physical Fitness on Cognitive Domains among Older Adults: A Cross-Sectional Study. Healthcare 2023, 11, doi:10.3390/healthcare11222963.
  3. Rezaee, M.; Hedayati, A.; Naghizadeh, M.M.; Farjam, M.; Sabet, H.R.; Paknahad, M. Correlation between Happiness and Depression According to Beck Depression and Oxford Happiness Inventory among University Students. Galen Medical Journal 2016, 5, 75-81, doi:10.31661/gmj.v5i2.598.
  4. Müller, M.; Turner, D.; Barra, S.; Rösler, M.; Retz, W. ADHD and associated psychopathology in older adults in a German community sample. Journal of Neural Transmission 2023, 130, 313-323, doi:10.1007/s00702-022-02584-4.

Reviewer 2 Report

Comments and Suggestions for Authors

This athors propose that the effects of exercise and nutrition on cognitive status are moderated and  partially mediated by happiness. This study is well designed. Only a few minor comments are listed in the attached pdf.  

Author Response

For research article

Response to Reviewer X Comments

1. Summary

Thank you very much for taking the time to review this manuscript. Please find the detailed responses below and the corresponding revisions highlighted in green in the re-submitted files.

2. Questions for General Evaluation

Reviewer’s Evaluation

Response and Revisions

Does the introduction provide sufficient background and include all relevant references?

Yes

Thank you.

Are all the cited references relevant to the research?

Yes

Thank you.

Is the research design appropriate?

Yes

Thank you.

Are the methods adequately described?

Yes

Thank you.

Are the results clearly presented?

Yes

Thank you.

Are the conclusions supported by the results?

Can be improved

Please see below.

3. Point-by-point response to Comments and Suggestions for Authors

This authors propose that the effects of exercise and nutrition on cognitive status are moderated and partially mediated by happiness. This study is well designed. Only a few minor comments are listed in the attached pdf. 

Comments 1: Page 2 lines 83-84, “who had no OASR record”, Define OASR here.

Response 1: Thank you for pointing this out. We have updated this to “who had incomplete cognition data” to leave the introduction of OASR in the Measures section 2.2.3, keeping consistent with other instruments.

Comments 2: Page 3 line 122, “in”, change to “in a”; line 127, “was”, change to “were”.

Response 2: Thank you for noting these. We have updated accordingly.

Comments 3: Page 8 lines 258-260, cite specific studies to support this claim. Otherwise, please reword to suggest that supplements may help.

Response 3: Thanks for your suggestion. We have both reworded the sentence and provided citation to support the claim. Please see page 8 lines 265-267 highlighted in green.

Reviewer 3 Report

Comments and Suggestions for Authors

In this manuscript Liu and colleagues present a work evaluating the nutritional status and physical activity are associated with cognitive function in chinese community-dwelling older adults.

The experiments and protocol are carried out properly and the finding presented and explained consequently, and their interpretation are solid. The manuscript by Liu and colleagues reports very interesting and new findings about this situation. The conclusions are convincingly supported by the data obtained, but some more detailed explanation must be included.

The introduction and the discussion do not provide a comprehensive and critical analysis of the existing literature. You should summarize, compare, and contrast the results presented with the findings of different studies and discuss their strengths, limitations and implications. It should also identify knowledge gaps and challenges.

Author Response

For research article

Response to Reviewer X Comments

1. Summary

Thank you very much for taking the time to review this manuscript. Please find the detailed responses below and the corresponding revisions highlighted in the re-submitted files.

2. Questions for General Evaluation

Reviewer’s Evaluation

Response and Revisions

Does the introduction provide sufficient background and include all relevant references?

Can be improved

We have improved the introduction.

Are all the cited references relevant to the research?

Yes

Thank you.

Is the research design appropriate?

Yes

Thank you.

Are the methods adequately described?

Yes

Thank you.

Are the results clearly presented?

Yes

Thank you.

Are the conclusions supported by the results?

Yes

Thank you.

3. Point-by-point response to Comments and Suggestions for Authors

In this manuscript Liu and colleagues present a work evaluating the nutritional status and physical activity are associated with cognitive function in chinese community-dwelling older adults.

The experiments and protocol are carried out properly and the finding presented and explained consequently, and their interpretation are solid. The manuscript by Liu and colleagues reports very interesting and new findings about this situation. The conclusions are convincingly supported by the data obtained, but some more detailed explanation must be included.

Comments 1: The introduction and the discussion do not provide a comprehensive and critical analysis of the existing literature. You should summarize, compare, and contrast the results presented with the findings of different studies and discuss their strengths, limitations and implications. It should also identify knowledge gaps and challenges.

Response 1: Thank you for pointing this out. We have improved the introduction to enhance its clarity and flow and discuss knowledge gaps and challenges. For discussion, we have substantially revised the contents according to your suggested flow, including summary of main findings, comparison with existing literature, potential mechanisms underlying findings, and finally implications and limitations. Please see highlighted contents in the introduction and discussion section for the updates.
